# Processing of an Audiobook in the Human Brain Is Shaped by Cultural Family Background

**DOI:** 10.3390/brainsci12050649

**Published:** 2022-05-15

**Authors:** Maria Hakonen, Arsi Ikäheimonen, Annika Hultèn, Janne Kauttonen, Miika Koskinen, Fa-Hsuan Lin, Anastasia Lowe, Mikko Sams, Iiro P. Jääskeläinen

**Affiliations:** 1Brain and Mind Laboratory, Department of Neuroscience and Biomedical Engineering, School of Science, Aalto University, 00076 Espoo, Finland; arsi.ikaheimonen@aalto.fi (A.I.); anastasia.lowe@aalto.fi (A.L.); mikko.sams@aalto.fi (M.S.); iiro.jaaskelainen@aalto.fi (I.P.J.); 2Department of Radiology, Massachusetts General Hospital, Harvard Medical School, Boston, MA 02114, USA; 3Faculty of Sport and Health Sciences, University of Jyväskylä, 40014 Jyväskylä, Finland; 4Advanced Magnetic Imaging Centre, School of Science, Aalto University, 00076 Espoo, Finland; 5Imaging Language, Department of Neuroscience and Biomedical Engineering, School of Science, Aalto University, 00076 Espoo, Finland; annika.hulten@gmail.com; 6Digital Business, Haaga-Helia University of Applied Sciences, 00520 Helsinki, Finland; janne.kauttonen@gmail.com; 7Faculty of Medicine, University of Helsinki, 00014 Helsinki, Finland; miika.koskinen@hus.fi; 8Sunnybrook Research Institute, Toronto, ON M4N 3M5, Canada; fhlin@sri.utoronto.ca; 9Department of Medical Biophysics, University of Toronto, Toronto, ON M5G 1L7, Canada; 10MAGICS Infrastructure, Aalto Studios, Aalto University, 02150 Espoo, Finland; 11International Social Neuroscience Laboratory, Institute of Cognitive Neuroscience, National Research University Higher School of Economics, 101000 Moscow, Russia

**Keywords:** narrative, cultural background, auditory, fMRI, inter-subject correlation

## Abstract

Perception of the same narrative can vary between individuals depending on a listener’s previous experiences. We studied whether and how cultural family background may shape the processing of an audiobook in the human brain. During functional magnetic resonance imaging (fMRI), 48 healthy volunteers from two different cultural family backgrounds listened to an audiobook depicting the intercultural social life of young adults with the respective cultural backgrounds. Shared cultural family background increased inter-subject correlation of hemodynamic activity in the left-hemispheric Heschl’s gyrus, insula, superior temporal gyrus, lingual gyrus and middle temporal gyrus, in the right-hemispheric lateral occipital and posterior cingulate cortices as well as in the bilateral middle temporal gyrus, middle occipital gyrus and precuneus. Thus, cultural family background is reflected in multiple areas of speech processing in the brain and may also modulate visual imagery. After neuroimaging, the participants listened to the narrative again and, after each passage, produced a list of words that had been on their minds when they heard the audiobook during neuroimaging. Cultural family background was reflected as semantic differences in these word lists as quantified by a word2vec-generated semantic model. Our findings may depict enhanced mutual understanding between persons who share similar cultural family backgrounds.

## 1. Introduction

People raised and living in one cultural milieu and speaking the same language largely share experiences, knowledge, beliefs, values, attitudes, and communication rules that together facilitate shared understanding and foster smooth cooperation. These similarities create “homophily”—love of the same—that has been studied in sociology for decades [1]. Neuroimaging evidence for homophily was provided only recently with findings showing that brain activity is more similar among friends—in contrast to acquaintances—when they are viewing audiovisual film clips [2]. Interest in the effects of cultural homophily on neural mechanisms of cognitive functions has increased rapidly, as exemplified by the emergence of the new field of “cultural neuroscience” [3,4]. Previous behavioral [5,6,7,8,9,10] and neuroimaging [11,12,13] studies have shown that persons living in Eastern (Asian) and Western (American) cultures exhibit within-culture similarities and between-culture differences in both lower-level processes, such as perception and number representation, and higher-order processes, such as contemplating the self and inferring others’ emotions (for reviews, see [3,12,13]). Such results have thus far been explained by cultural influences in perceptual-cognitive styles [3,4]. However, the effects of shared cultural family backgrounds within a modern multi-ethnic nation are less well-known than those arising in international/intercontinental comparative studies. A study that investigated the perceptual-cognitive styles of Asian Americans and European Americans found that the styles of Asian Americans were either identical to those of European Americans or fell in-between the styles of East Asians and European Americans, thus suggesting that perceptual-cognitive styles readily adjust to the cultural context that person lives [14]. Interestingly, it has also been found that Chinese students showed a significant shift in their perceptual-cognitive style after 6 months when they engaged in a course at a British university [15]. This suggests that cognitive styles can be affected by even subtle cultural influences.

Neuroimaging allows one to study how language is processed in multiple brain areas. Language processing includes prelexical processing, which involves the auditory cortices at superior temporal lobes, whereas individual words and short sentences are associated with left-hemisphere lateral and anterior temporal cortical activity [16]. In contrast, continuous speech stimulus results in more widespread and bilateral activity involving the prefrontal and parietal areas as well as the cingulate cortices and precuneus that have been associated with understanding storylines in narratives [17,18,19,20,21]. The visual or occipital cortex is also involved while listening to a narrative, possibly relating to visual imagery elicited by the narrative [22,23].

The influence of cultural background on language processing is still largely unknown. Previous cross-cultural behavioral studies have provided evidence that cultural background can shape the understanding and interpretation of written narratives [24,25]. For example, studies have found that there are cross-cultural differences in narrative comprehension, character evaluation, plot development, and time/space imagination [24,25]. It has also been shown that the N400 event-related potential response amplitude of fluent Welsh–English bilinguals was significantly stronger for sentences written in Welsh than for sentences written in English [26]. This difference was only found for sentences that contained information about Welsh culture but not for sentences that did not contain culture-related information. The influence of cultural background on language processing might be important given that in multicultural societies, native speakers of the same language may assume highly similar understandings of concepts [27], thus potentially causing mutual misunderstanding.

Recent methodological advances have made it possible to both record brain activity [2,17,19,20,23,28,29,30,31] and behaviorally assess subjective interpretations [22] while listening to an audiobook. A widely used method for analyzing fMRI data acquired in naturalistic paradigms is intersubject correlation (ISC) analysis where voxel-by-voxel correlations are computed between the hemodynamic activity of each pair of participants [32]. A major benefit of ISC analysis is that it is model-free in that it does not require a regression model since the hemodynamic activity of a reference subject is used to predict the hemodynamic activity of another subject. Recently developed statistical methods also allow for the estimation of semantic similarity between words that participants have reported when they have been asked to freely recall a movie [30] or describe associations elicited by a narrative [22]. Word2vec is an advanced statistical model that can be used to transform words into vectors [33]. Semantic similarity between words can then be evaluated by computing cosine similarity between those vectors.

Here, we utilized ISC and word2vec to investigate the influence of cultural family background on interpretation and processing of an audiobook in the human brain. Based on previous studies that have found culture-specific experiences to shape perception and cognition [3,4,24,34], we hypothesized that word2vec reveals cultural family-background-specific enhancements of similarity in how different parts of the story are interpreted. Further, we expected that similar interpretations of the narrative result in higher similarity of brain hemodynamic activity in brain areas known to be involved in the processing of natural speech. These results will extend previous international/intercontinental comparative studies [3,4,24,34] by showing that even slight differences in cultural background within a modern multi-ethnic nation can modify perception. An audiobook as a stimulus allowed us to investigate neural dynamics in closer-to-real-life settings compared to previous studies in the field of cultural neuroscience that have typically used simple, controlled paradigms [3,4,24,34].

## 2. Materials and Methods

### 2.1. Participants

Forty-eight right-handed (Edinburgh Handedness Inventory, [35]) participants participated in the study. Among them, 24 participants had a Finnish family background and 24 participants had a Russian family background. The participants with the Finnish family background had Finnish parents and were born and raised in Finland. They were aged between 19 and 35 years (mean age: 24.7 years). For 12 of the Russian-background participants, both parents were Russian; of the remaining participants, 11 had a Russian mother and one had a Russian father. The participants with Russian family background were aged between 18 and 35 years (mean age: 24.7 years). Twenty of the Russian-background participants were born in Finland; one participant had moved to Finland at the age of eight, one at the age of one, and two at the age of three years. Ten of the participants with the Russian family background were exposed to both Finnish and Russian in the first years of their lives, thirteen started to acquire Finnish at the age of three years when they went to preschool, and one started to acquire Finnish at the age of eight years. The Finnish fluency of the Russian-background participants was evaluated with questions in Appendix B. Further, during the recruitment and experimental procedures, Finnish fluency of all participants was first-hand confirmed as all communications took place in Finnish. Finnish fluency would have been measured more accurately with a language test and more detailed questionnaires, but these were not used to keep the duration of the experiment bearable to the participants. All participants were Finnish residents (11 Russian background participants also had Russian citizenship) and had graduated from a Finnish primary school. Half of the participants in both groups were females. In both groups, one of the participants had a vocational degree and the others had a Matriculation, undergraduate or university degree. None of the participants self-reported any neurological or hearing deficits. Prior to participation, all participants gave their written voluntary informed consent. The study was approved by the Aalto University Research Ethics Committee and conducted in accordance with the Declaration of Helsinki. Note that the participants did not constitute a representative sample of Finnish or Russian family background residents of Finland, and the results can therefore not be generalized to the population level.

It should be noted that the goal of this study was not to investigate the Finnish- and Russian-background populations in Finland per se. Rather, the current study is basic research that intends to answer the intriguing question of whether family cultural background can influence how continuous speech is perceived and processed in the brain. Thus, the participants were not recruited with population-sampling methods, the groups are not representative of Finnish- and Russian-background populations in Finland, and our results cannot be generalized to these populations. For example, the gender, age, and education of the participants are not representative of these populations. Instead, these factors were matched between the sample groups so that they did not confound the results of between-group comparisons.

### 2.2. Stimulus

During the fMRI measurements, the participants were presented with a 71 min fictional audiobook in Finnish (custom written by author IPJ; Appendix A: the English-translated textual version). The audiobook was presented during ten measurements (durations: 4.8–8.4 min) between which the participant had a short break in the scanner. The narrative included episodes related to and occasionally contrasting Finnish and Russian cultures (e.g., religion, food, literature, history, and traditions). The protagonists of the narrative are two young adult friends living in Helsinki (Finland) who have a Finnish or a Russian background, respectively. In short, the Finnish man is dating a Russian-background woman, whereas the Russian man is in a relationship with a Finnish woman. The characters are described as having personality and behavioral characteristics stereotypically associated with Finns or Russians (e.g., Russian characters were more emotional and conservative than Finns), but without exaggeration to present the protagonists as realistic. Cultural differences in expressing emotions, style of thinking, behavioral patterns, values, and attitudes result in both difficulties in understanding each other and in conflict between the characters. At the end of the story, the Russian-background man ends up in a relationship with the Russian-background woman and the Finnish man with the Finnish woman. The aim of the story was that the Finnish-only background participants would identify more with the Finnish protagonists and understand the Finnish culture-specific elements, while the participants with a Russian background would identify more with the Russian protagonists and have greater understanding with the Russian-specific elements. The periods of the narrative describing interactions between protagonists were interspersed with periods without social interactions or culture-specific elements. The audiobook was recorded with a sampling frequency of 44,100 Hz in a professional recording studio. For the fMRI experiment, the auditory file of the audiobook was divided into 10 segments and, for the free association task, into 101 segments.

### 2.3. Experimental Procedures

The participants filled out the background questionnaire (Table 1) as part of the behavioral questionnaires. Thereafter, participants engaged in the fMRI measurement and a magnetoencephalography (MEG) measurement with concurrent electroencephalography (EEG). The brain measurement sessions were separated by at least one month to reduce possible learning effects. The order of the neuroimaging sessions was counterbalanced across the cultural family backgrounds and genders of the participants. MEG/EEG results will be reported separately.

After the brain measurements, the participants performed the implicit association test (IAT) which aims to measure automatic associations or beliefs that the participant is not aware of or willing to report [36,37]. However, given that the IAT is controversial [38,39,40], we reported IAT results only in the appendixes as tentative support of the main results of this study and our conclusions (see Appendix C and Appendix E).

The participants also filled out an online free association task at home within two weeks of neuroimaging. In this task, the audiobook was presented once again to the participant in 101 segments, and at the end of each segment, the participants were instructed to type a few words that best describe what was on their mind at the end of the segment when they had heard that segment during brain measurements. The segments are indicated in the English-translated version of the audiobook in the Appendix A. Boosted decision tree regression was used to study which of the background variables most accurately predicted the results of the free association experiment. The details and results of this analysis are presented in Appendix D and Appendix F.

### 2.4. fMRI Acquisition

The fMRI session consisted of the following 13 measurements: (1) a T1-weighted anatomical measurement, (2) a measurement comprising four repetitions of the introduction of the narrative (5.93 min), (3) the narrative presented in ten measurements (durations: 4.8–8.4 min), (4) a measurement comprising four repetitions of the introduction of the narrative (5.93 min), (5) a resting state measurement (3 min) and (6) a T2-weighted anatomical measurement. In this study, we analyzed only the narrative (i.e., ten fMRI runs) and T1-weighted anatomical images. Each functional measurement started with a period of 12.3 s and ended with a period of 15 s without auditory stimulation. The repetitions of the introduction were separated by a period of 16 s without auditory stimulation. The participant was shown a white fixation cross on a black background during the functional measurements (including resting state). Images from Helsinki were shown between the functional measurements and during the anatomical measurements. The participants had short breaks in the scanner between the measurements to avoid fatigue, eye strain, auditory strain from the repetitive noise of the scanning, and muscle strain from laying still. We also inquired about their fatigue between the measurements and offered a possibility to have a break outside of the scanner, during which they were offered refreshments. Among them, 24 participants wanted to have 1, 6 participants 2–4, and 18 participants 0 breaks outside of the scanner between measurements.

Anatomical and functional MRI data were acquired with a 3T MRI whole-body scanner (MAGNETOM Skyra, Siemens Healthcare, Erlangen, Germany) using a 32-channel receiving head coil array. The anatomical images were measured using a T1-weighted MPRAGE sequence (TR = 2530 ms, TE = 3.30 ms, field of view = 256 mm, flip angle = 7 degrees, slice thickness = 1 mm). Whole-brain fMRI data were measured using an ultra-fast simultaneous multislice (SMS) inverse imaging (InI) sequence [41]. Instead of relying only on gradient coils in spatial encoding, InI achieves spatial encoding by solving the inverse problems by utilizing the spatial information from channels in a radio-frequency coil array and gradient coils. In this study, InI-encoding direction was superior–inferior, whereas frequency and phase encoding were used to recover spatial information in anterior–posterior and left–right directions, respectively. Twenty-four axial slices (7 mm) were first collected without a gap between the slices. Thereafter, the slices were divided into two groups of 12 slices, and each of the slice groups was excited and read in 50 ms, resulting in a TR of 100 ms. Simultaneous echo refocusing [42] was used to separate adjacent slices in each group, and aliasing was further controlled with blipped controlled aliasing in parallel imaging [43]. Other measurement parameters were: TE = 27.5 ms, flip angle = 30°, FOV = 210 × 210 × 210 mm^3^, and in-plane resolution = 5 mm × 5 mm.

Solving an inverse problem in InI reconstruction requires a sensitivity map of the channels in the coil array [41]. This information was included in a 6 s reference scan measured before each functional measurement. In this reference scan, partition-encoding steps were added after slice group excitation in an InI-encoding direction. The reference scan and functional scans were acquired with the same imaging parameters. Before each reference scan, shimming was used to minimize inhomogeneity in the magnetic field.

SMSInI should, due to its faster sampling rate, enable more accurate removal of physiological artifacts compared to the echo planar imaging (EPI) sequence [44]. SMSInI has higher spatial resolution with lower signal leakage and higher time-domain signal-to-noise ratio than inverse imaging without SMS, and it detects subcortical fMRI signals with similar sensitivity and localization accuracy as EPI [41]. The in-plane resolution of about 5 × 5 of SMSInI is lower compared to the resolution of about 3 × 3 mm of the EPI sequence. However, EPI data are usually smoothed spatially and, therefore, the eventual spatial resolution is typically closer to 5 × 5 mm. Further, recent studies have provided evidence that fMRI can measure faster neural oscillations than previously thought [45,46]. This evidence has challenged the assumption that decreasing TR does not provide any additional neural information since the hemodynamic response is too slow to measure neural oscillations.

During the fMRI measurements, the stimulus presentation was controlled with the Presentation software (Neurobehavioral Systems, ver. 18.1, Albany, NY, USA). The auditory stimuli were presented to the participant through earphones (Klaus A. Riederer ADU2a). The intensity level of the stimuli was adjusted on an individual participant basis at a comfortable listening level that was clearly audible above the scanner noise. The audio out of the sound card of a computer was recorded with the BIOPAC MP150 Acquisition System (BIOPAC System, Inc., Goleta, CA, USA). In the data analysis, this allowed us to determine the exact times when the auditory stimulus started. The BIOPAC MP150 system was also used to record heart rate and respiration signals during the fMRI measurement. Heart rate was measured using two BIOPAC TSD200 pulse plethysmogram transducers placed on the palmar surfaces of the participant’s left and right index fingers. Respiratory movements were measured using a respiratory-effort BIOPAC TSD201 transducer attached to an elastic respiratory belt, which was placed around the participant’s chest. Heart rate and respiratory signals were sampled simultaneously at 1 kHz using RSO100C and PPG100C amplifiers, respectively, and BIOPAC AcqKnowledge software (version 4.1.1).

### 2.5. fMRI Reconstruction and Preprocessing

The anatomical images were reconstructed using Freesurfer’s automatic reconstruction tool (recon-all; http://surfer.nmr.mgh.harvard.edu/, accessed on 23 August 2018) and functional images using the regularized sensitivity encoding (SENSE) algorithm with a regularization parameter of 0.005 [47,48]. The reconstructed images were registered to the Montreal Neurological Institute 152 (MNI152) standard space template by first calculating transformation parameters from structural to standard space and from the reference scan to structural space. Thereafter, these transformations were concatenated and used to co-register functional images to the MNI152 standard space with 3 mm resolution. The co-registrations were performed by Freesurfer and FSL tools [49,50]. A period of 12.3 s of fMRI data measured before the start of the audiobook was removed from each fMRI measurement. To remove the scanner drift, the data were detrended using a Savitzky–Golay filter (order: 3, frame length: 240 s). Physiological and movement artifacts were suppressed using the MaxCorr method [51]. Specifically, from the data of each participant, we regressed out ten components that correlated maximally within the white matter and cerebrospinal fluid of that participant but were minimal in the other participants’ white matter and cerebrospinal fluid. Since these components were participant-specific, they were assumed to be artifacts rather than brain activity elicited by the audiobook. Thereafter, DVARS, defined as the spatial standard deviation of successive difference images, was used to identify the data potentially affected by head motions [52]. The fMRI data were filtered between 0.08 and 4 Hz using a zero-phase filter and smoothed spatially with a 6 mm full-width–half-maximum Gaussian kernel.

### 2.6. Inter-Subject Correlation (ISC) Analysis of Blood-Oxygen-Level-Dependent Responses

Blood-oxygen-level-dependent (BOLD) responses were analyzed using ISC analysis as implemented in the ISC toolbox (https://www.nitrc.org/projects/isc-toolbox/ [53], accessed on 23 November 2017). First, we investigated how similar brain activity was across the whole audiobook within Finnish- and Russian-background groups by computing ISC maps separately for both groups. For each voxel and for each of the 10 functional measurements, Pearson’s correlation coefficients were calculated across the time courses of every participant pair within the Finnish- and Russian-background groups, resulting in 276 (*n* × (*n* – 1)/2, where *n* = 24) ISC values per voxel for each group. To calculate the ISC values across the whole audiobook, the ISC values for each measurement were transformed into z-values using Fisher’s transformation, the z-values were weighted with the length of the measurement and, thereafter, an average was calculated over the z-values within each voxel across 10 measurements. Finally, the z-values were transformed back into ISC values using the inverse z-transform.

The statistical significance of the ISC values was evaluated separately for the Finnish- and Russian-background groups by a nonparametric voxel-wise resampling test to account for the temporal autocorrelations in the BOLD data [53]. In short, a null distribution was created from ISC values calculated after circularly shifting the time series for each participant by a random amount such that the time series between participants became unaligned in time. The null distribution was approximated with 0.5 million realizations randomized across voxels and time-points. The null distribution across the whole audiobook was determined by taking a weighted average over the null distributions computed for ten functional measurements. Again, the ISC values of the null distribution were transformed into z-values before averaging and the averaged z-values were transformed back into the ISC values using the inverse z-transformation. The threshold for significant ISC was determined by first computing *p*-values for the true realizations (i.e., the ISC values computed with non-shifted BOLD timeseries) in each voxel based on the null distribution and, thereafter, the resulting *p*-values were corrected for multiple comparisons using the FDR correction.

Between-group differences of Fisher’s z-transformed ISC values (z-scores) were studied using a nonparametric cluster-based two-sample *t*-test where the statistical significance was determined based on the distribution of 5000 random permutations of z-values between the groups [54,55,56,57,58]. The null distribution was created from the maximum cluster sizes obtained by thresholding the statistical images at the cluster-defining threshold of *p* < 0.001 at each permutation. A corrected *p*-value for each suprathreshold cluster was obtained by comparing its size to the permutation distribution.

### 2.7. Semantic Similarity Analysis of the Self-Reported Word Lists

To examine between-group differences in word lists produced by the participants in the free association task, the word2vec skip-gram method [33] was used to transform the individual words into vectors representing the semantic content. The word2vec skip-gram model (Gensim Python Library, https://pypi.org/project/gensim/, accessed on 16 July 2018) was trained using the Finnish Internet Parsebank corpus ([59], http://bionlp.utu.fi/finnish-internet-parsebank.html, accessed on 11 May 2018) with 500 dimensions and a window size of 10 words. The words that occurred less than 50 times in the corpus were excluded from the model training. The vector embeddings were adjusted by passing the corpus through the word2vec skip-gram model five times.

The model contained 98% of the words produced by the participants, and the rest of the words were discarded. The word lists produced by the participants were first corrected for spelling errors and stop-words were removed in the case the participant had written sentences. Thereafter, word2vec was used to map each word to a semantic vector. The semantic representation of each list was obtained by calculating the vector sum over the words in the list. The list vector was computed for 101 narrative segments. Between-participant pairwise semantic similarities were then obtained by calculating cosine distances between the list vectors separately within the Finnish- and Russian-background groups for each narrative segment (i.e., 101 *t*-tests).

The pairwise cosine similarities within the Finnish-background participants were compared to the pairwise cosine similarities between Russian-background participants using a nonparametric two-tailed *t*-test with 50,000 permutations [54,56,57,58]. The *t*-tests were performed separately for each of the 101 segments. The segment-wise *p*-values (i.e., 101 in total) were corrected for multiple comparisons using FDR correction [60]. We also applied the two-tailed Wilcoxon signed rank test to investigate whether the number of associated words across the whole narrative (i.e., one statistical test) differed between the Finnish- and Russian-background participants.

## 3. Results

### 3.1. Audiobook Listening Elicited Widespread ISC of BOLD Activity

There was significant ISC in several brain regions during audiobook listening (Figure 1 and Table 2; activation maps can be found from Neurovault: https://neurovault.org/collections/LINKKSTF/). Areas of ISC shared across both groups included Heschl’s gyrus (HG), planum temporale (PTe), as well as superior and middle temporal gyri (STG, MTG) bilaterally, further extending to lateral aspects of occipital cortices. The Finnish-background participants exhibited more extended ISC in the lateral occipital cortex (LOC), especially in the right hemisphere and also the inferior occipitotemporal cortex (iOTC). Significant ISC was also found in both groups in the left inferior frontal gyrus (IFG) and bilateral middle frontal gyri (MFG), precuneus (PCun) and anterior cingulate cortex (ACC).

### 3.2. Cultural Family Background Modulates Brain Activity during Audiobook Listening

Stronger ISC was found for the Finnish rather than the Russian background participants in the left hemisphere in an area extending from the Heschl’s gyrus and insula to the superior temporal gyrus, as well as in an area extending from the lingual gyrus to the middle occipital gyrus and cerebellum (Figure 2, cerebellum not shown; Table 3). In the right hemisphere, ISC was stronger for the Finnish-background participants in an area including parts of the middle temporal and middle occipital gyri, lateral occipital and posterior cingulate cortices, as well as in the cerebellum. The ISC was stronger for the Russian compared to the Finnish -background participants in left-hemisphere areas extending from Heschl’s gyrus to the superior and middle temporal gyri, as well as bilaterally in posterior–inferior parts of the precuneus and anterior parts of the cuneus. No differences were found in DVARS values between the two participant groups, suggesting that the between-group differences found in the ISC values should not be related to the head movements (Finnish family background: 1.64 ± 0.46, Russian family background: 1.59 ± 0.41, *t*-value: 0.70, *p* = n.s., the nonparametric two-tailed *t*-test with 5000 permutations).

### 3.3. Cultural Family Background Increased Similarity of Audiobook Interpretation

We estimated similarities in interpretation of the audiobook using the free association task. Participants listened to the audiobook in 101 segments, and after each segment, they typed a list of words that described what had been on their mind at that point in the story during the brain measurement session (Figure 3; for a description of the method, see also [22]). The semantic similarity of the participants’ word lists was estimated by transforming the lists into vector representations in a semantic space and by calculating the cosine similarities between the resulting semantic vectors [33]. Across the whole narrative, the Russian-background participants listed 33% more words than the Finnish-background participants (18,724 vs. 12,536 words; the two-tailed Wilcoxon signed rank test: V = 49, *p* < 0.01). Similarities of provided words were significantly different between groups in 56/101 segments (*p* < 0.05 in each segment, as assessed with the two-tailed nonparametric *t*-test with FDR adjustment). Out of these 56 segments, similarities were higher for the Russian-background participants in 44 segments and lower for 12 segments. There were significant between-group differences both in segments depicting interacting protagonists and in segments with descriptions of city scenery amid changing seasons. The segments with significant between-group differences are indicated in the English-translated transcription of the audiobook in the Appendix A.

## 4. Discussion

In the present study, we were able to show that differences in cultural family background can be reflected as between-group differences in ISC of brain hemodynamic activity during audiobook listening in several brain areas, even when participants of both groups are fluent in Finnish (see, Section 2.1) and are Finnish residents. To our knowledge, this is the first time such an effect has been found. This is an important extension of the previous studies, the majority of which have found cross-cultural differences between residents of different continents or countries (usually Americans vs. East Asians; for reviews, see [3,4,12,13]). The audiobook allowed a more natural setting to investigate the effect of cultural background than previously used simple and strictly controlled stimuli. ISC provided complementary insights to traditional analyses since it does not require a prior model of brain responses to the stimulus and is, therefore, especially suitable for analyzing brain responses to complex naturalistic stimuli. Here, between-group differences in ISC were found in the left-hemispheric Heschl’s gyrus, insula, superior temporal gyrus and lingual gyrus, right-hemispheric lateral occipital and posterior cingulate cortices as well as bilateral middle temporal gyrus, middle occipital gyrus, and precuneus (Figure 2). These findings suggest that a shared cultural family background enhances similarity of processing of an audiobook at multiple levels of language processing.

The left superior temporal gyrus, the bilateral middle temporal gyrus, and the precuneus have previously, among other things, been associated with narrative speech comprehension and processing of semantic information ([63]; for reviews, see [64,65]). Further, the precuneus exhibited increased ISC between participants who understood visual and auditory narratives more similarly [30]. The precuneus is also involved, for example, in the processing of paragraphs about in-group vs. out-group characters [66] and in episodic memory retrieval [67]. In light of these previous studies, the between-group ISC differences in our study seem to suggest that cultural family background can shape semantic processing. The between-group ISC differences in the left Heschl’s gyrus and the surrounding areas, on the other hand, seem to suggest that cultural family background may also modulate the processing of the speech acoustics and prelexical features [16,64,68]. Here, top-down influences from the higher-level speech processing areas may play an important role [69,70,71]. Differences in ISC in Heschl’s gyrus and the higher-level associative brain areas support previous studies that have found cultural differences in both lower-level (e.g., perception, attention) and higher-level processes (e.g., inferring other’s emotions, contemplating the self [3,11,12,13,34,72]). Interestingly, between-group differences in the ISC were also found in the occipital cortex. Given that visual imagery may activate the same areas as visual perception [73,74,75], the present results could reflect an effect of the cultural family background on the visual imagery elicited by the audiobook narrative.

Using word2vec and semantic similarity, we were able to quantitatively evaluate the closeness of meaning of the word associations elicited by the audiobook. The behavioral word-listing experiment disclosed that 56 out of the 101 audiobook segments elicited significantly different associations between the participants with Finnish and Russian family backgrounds (Figure 3). The semantic similarity of the associated words was higher among the Russian-background participants in 44 segments of the story compared to the Finnish-background participants, and vice versa in 12 segments. This suggests that those story segments contained elements that elicited cultural/family background-related associations specific to each group and may have been understood more similarly within each group. These segments contained both culture-specific but also non-specific elements (see the English-translated text of the audiobook in the Appendix A) suggesting that the cultural family background may also modify the way we understand concepts that are not obviously related to the culture. This latter observation may stem from cultural differences in perceptual styles as described in previous studies on object perception [11], color discrimination [76], and taste [77]. Our results are also in line with studies that have found cross-cultural differences in reading comprehension and interpretation [24,25]. For example, it has been shown that when asked to recall culturally unfamiliar texts, readers tend to distort facts and insert ideas from their own culture [78]. Overall, the Russian-background participants produced 33% more words than the Finnish-only background participants, which could be explained by the Finno-Ugric culture having been observed to be a less talkative one [79].

This study has some limitations. First, the analyses employed in this study do not demonstrate a causal relationship between cultural background of the participants and narrative processing and interpretation. The between-group differences may be related to differences in cognitive or emotional demands during narrative listening or to viewing the story protagonists as belonging to the participant’s own group vs. another cultural group. Therefore, further studies should be conducted to validate and extend these initial results that suggest that cultural background can modulate narrative processing and interpretation. For example, a follow-up study where immigrants are studied before and after immigration to another cultural environment might offer a stronger basis for inference of the causal relationship of culture, in the case of cultural immersion. Second, in the free association test, the participants were instructed to type words that were on their mind while listening to the story during neuroimaging, but it is still possible that the participants reported words that were on their mind only during the free association task. However, this was not a problem in the current study, since the fMRI data and associated words were analyzed separately. Furthermore, previous research has shown that the stimulus can serve as an effective recall cue of what participants experienced during fMRI scanning [80]. Third, having breaks outside of the scanner reduces comparability between measurements within the participant since the position of the participant’s head is changed when the participant is repositioned in the scanner. However, it is unlikely that this has significantly affected our results, since ISC values were first computed separately for each measurement over which an average ISC was computed. Fourth, the sections of the audiobook that included elements from either Finnish or Russian cultures were created based on interviews with Russian-background persons. A more objective method to assess whether the sections are related more to Finnish or Russian cultures would be to ask independent raters to categorize the sections. Still, many of the Russian-background participants relayed that the stimulus realistically captured aspects of Russian culture, and there were significant differences in the free association task between the Finnish- and Russian-background participants, so here the stimulus did work as desired. Finally, to see whether the SMSInI sequence provided benefit in this study, the results should be compared to EPI data collected from the same experiment. In theory, a 100 ms TR used in this study should allow more accurate removal of physiological artifacts than the conventional 2–3 s TR of EPI, which is too low to avoid aliasing of higher frequency cardiac and respiratory cycles. However, SMSInI also introduces additional artifacts, such as susceptibility artifacts. Further, it is unclear whether fast sampling can capture higher-frequency information of neuronal activity, since the hemodynamic response is inherently slow, although there is evidence that fMRI can measure neural oscillations as fast as 0.75 Hz [45,46].

In conclusion, our results suggest that even relatively subtle between-group differences in cultural background can result in higher similarity within one group compared to the other in how natural speech is interpreted and how it is processed in multiple areas in the human brain. These effects might play an important role in enhancing mutual understanding between individuals with shared cultural backgrounds. Awareness of these effects could help overcome potential challenges in mutual understanding between individuals with different cultural backgrounds and social identities in modern multicultural societies wherein individuals from different backgrounds speak the same native language and might erroneously take highly similar understanding for granted. Enhanced awareness and scientific understanding of these differences in language use is important for avoiding obstacles in social interactions at many levels of society.

## Figures and Tables

**Figure 1 brainsci-12-00649-f001:**
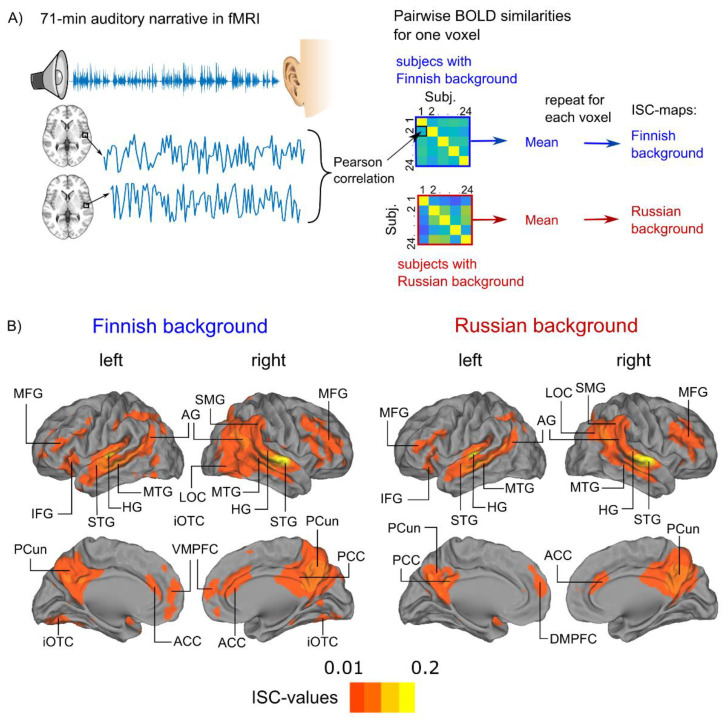
**Intersubject correlation** (ISC) during narrative listening. (**A**) Schematic illustration of the experimental paradigm and ISC analysis (see Methods for details). (**B**) Significant ISC of hemodynamic activity displayed on cortical surfaces at a voxel-wise false-discovery rate threshold of 0.001. Abbreviations: fMRI = functional magnetic resonance imaging, MFG = middle frontal gyrus, AG = angular gyrus, IFG = inferior frontal gyrus, STG = superior temporal gyrus, HG = Heschl’s gyrus, MTG = middle temporal gyrus, PCun = precuneus, iOTC = inferior occipitotemporal cortex, LOC = lateral occipital cortex, VMPFC = ventromedial prefrontal cortex, ACC = anterior cingulate cortex, PCC = posterior temporal cortex, DMPFC = dorsomedial prefrontal cortex. The loci of ISC were labelled according to the Harvard–Oxford Cortical Structural Atlas implemented in the FMRIB Software Library (FSL, https://fsl.fmrib.ox.ac.uk/fsl/fslwiki, accessed on 12 May 2022).

**Figure 2 brainsci-12-00649-f002:**
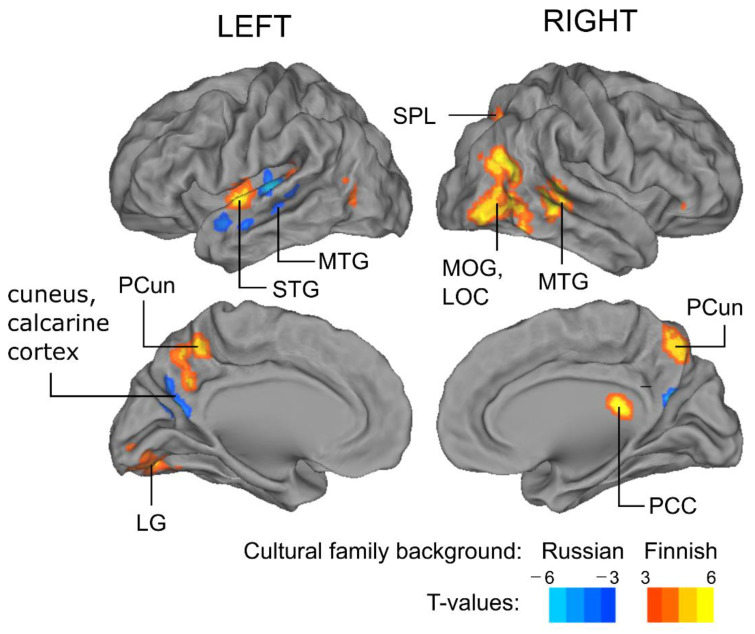
Brain areas where the activity was shaped by cultural family background. The brain areas where ISC was significantly different between the participants with the Finnish vs. Russian family backgrounds (*p* < 0.05, adjusted with cluster-based thresholding using 5000 permutations and cluster-defining threshold of *p* < 0.001, uncorrected). Red–yellow means stronger ISC for the Finnish background and blue–light blue stronger ISC for the Russian background participants. Abbreviations: STG = superior temporal gyrus, MTG = middle temporal gyrus, LOC = lateral occipital cortex, MOG = middle occipital gyrus, PCun = precuneus, SPL = superior parietal lobule, LG = lingual gyrus, PCC = posterior cingulate cortex. The loci of ISC were labeled according to the Harvard–Oxford Cortical Structural Atlas [61] implemented in FSL. The results were visualized on the cortical surface using Caret software [62]. Please note that the ISC difference in PCC was misleadingly projected in the ventricle when visualized using Caret.

**Figure 3 brainsci-12-00649-f003:**
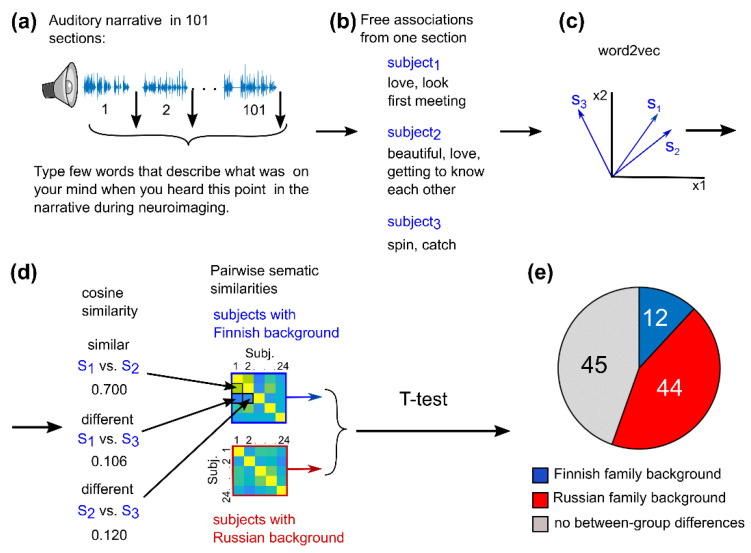
Schematic illustration and results of the behavioral free association task. (**a**) In the free association task, the audiobook was presented to the participant in 101 segments after the fMRI session. (**b**) After each segment, the participant typed in 20–30 s words that described what was on his/her mind at that point of the audiobook during fMRI. (**c**) The associated words were transformed into 500-dimensional vector representations in a semantic space (word2vec). For each segment, a vector sum was calculated over the vector representations of the words the participant had produced. (**d**) Thereafter, pairwise cosine similarities were calculated between these semantic vectors. Then, t-statistics was used to examine whether the cosine similarities differed between the word lists produced by the two groups of participants. (**e**) The Finnish-background participants produced semantically more similar word lists in 12 out of 101 segments; the Russian-background participants listed semantically more similar words in 44 out of 101 segments. There were no significant effects in 45 segments.

**Table 1 brainsci-12-00649-t001:** Results from the background questionnaire.

Cultural Background	Level of Command of Russian Language ***	Lifetime Visits to Russia ***	How Finnish Do I Feel Myself? (Scale 1–100) **	How Russian Do I Feel Myself? (Scale 1–100) ***	How Positively/Negatively Do I See Finns (Scale 1–100) n.s.	How Positively/Negatively Do I See Russians (Scale 1–100) n.s.
Finnish background	1.2 ± 0.1	1.4 ± 0.1	93.3 ± 1.9	1.6 ± 0.3	86.3 ± 2.2	59.7 ± 2.7
Russian	4.7 ± 0.1	3.9 ± 0.1	69.6 ± 3.1	55.8 ± 3.2	77.3 ± 2.6	67.3 ± 2.8

Results are mean values ± SEM. Level of command of Russian language: 1 = I don’t speak Russian, 2 = poor, 3 = satisfactory, 4 = good, 5 = native. Visits to Russia: 1. never visited, 2. visited few times, 3. visited several times, 4. visited regularly, 5. lived longer time periods. How Finnish I feel myself: 0 = not at all, 100 = strongly. How Russian I feel myself: 0 = not at all, 100 = strongly. How positively/negatively I see Finns: 0 = very negatively, 100 = very positively. How positively/negatively I see Russians: 0 = very negatively, 100 = very positively. Significance levels of between-group differences: nonsignificant (n.s.) > 0.05, ** *p* < 0.01, *** *p*< 0.001, tested with 50,000 permutations.

**Table 2 brainsci-12-00649-t002:** ISC values across Finnish- and Russian-family-background participants. Labels, sizes, peak coordinates, and maximum intersubject correlation (ISC) values of clusters obtained in the ISC analysis for the participants with Finnish and for the participants with Russian family backgrounds. The loci of ISC were labeled according to the Harvard–Oxford Cortical Structural Atlas implemented in the FMRIB Software Library (FSL, https://fsl.fmrib.ox.ac.uk/fsl/fslwiki, accessed on 12 May 2022). The coordinates are in Montreal Neurological Institute and Hospital (MNI) coordinate space. L = left, R = right.

Cluster Label	Cluster Extent (Voxels)	x MNI (mm)	y MNI (mm)	z MNI (mm)	Max ISC Value
**ISC Finnish background**					
Planum temporale (R)	1730	60	−12	0	0.14
Heschl’s gyrus (L)	1568	−48	−18	3	0.14
Frontal Orbital cortex (L)	82	−39	27	−9	0.02
Frontal pole (R)	69	48	42	3	0.02
Precuneus cortex (R)	1338	6	−63	33	0.06
Frontal pole (L)	68	−36	36	9	0.02
Cingulate gyrus (R)	71	6	33	21	0.02
**ISC Russian background**					
Cerebellar crus II (R)	38	30	−75	−39	0.04
Superior temporal gyrus, posterior division (R)	1055	63	−12	0	0.14
Heschl’s gyrus (L)	880	−48	−21	3	0.16
Inferior frontal gyrus (R)	72	45	18	24	0.03
Precuneus cortex (R)	920	6	−63	30	0.06

**Table 3 brainsci-12-00649-t003:** Differences of ISC values between Finnish- and Russian-family-background participants. Cluster size, peak coordinates, the maximum *t* -values and *p*-values of the clusters obtained in the nonparametric two-sample *t*-test between the two participant groups (5000 cluster-wise permutations; cluster-defining threshold, uncorrected: *p* < 0.001; the 0.05 family-wise error-corrected cluster size: 62 voxels). The loci of ISC were labeled according to the Harvard–Oxford Cortical Structural Atlas [61] implemented in FSL.

Cluster Label	Cluster Extent (Voxels)	x MNI (mm)	y MNI (mm)	z MNI (mm)	Max *t*-Value	*p*-Value
**ISC Finnish > ISC Russian**						
Temporal pole (superior, R)	87	54	21	−24	6.79	*p* < 0.03
Middle occipital gyrus (L)	468	−42	−66	3	6.81	*p* < 0.002
Middle temporal gyrus (R)	481	57	−63	12	6.92	*p* < 0.002
Middle temporal gyrus (R)	143	51	−42	3	5.89	*p* < 0.02
Superior temporal gyrus (L)	217	−57	−6	0	6.32	*p* < 0.01
Precuneus cortex (R)	401	18	−60	33	7.74	*p* < 0.002
**ISC Russian > ISC Finnish**						
Superior temporal gyrus (L)	210	−48	−24	6	−6.25	*p* < 0.01
Calcarine cortex (L)	85	−9	−69	18	−5.23	*p* < 0.03

## Data Availability

Stimulus material and codes used in the current study are available from the corresponding author on reasonable request. Pseudonymized fMRI are available within the European Union (EU) from the corresponding author by reasonable request in respect of the privacy of the participants following the guidelines of the Data Protection Act of Finland (includes the EU’s General Data Protection Regulation, GDPR). Since the facial features of the anatomical T1 MRI images are needed in the analysis of magnetoencephalography data also measured in this study, T1 images will be available within the EU by reasonable request following the guidelines of the Data Protection Act of Finland only after the end of the project when facial features of the MRI volumes can be removed.

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
