# Peer review of "Processing of an Audiobook in the Human Brain Is Shaped by Cultural Family Background"

_brainsci, 2022, doi:10.3390/brainsci12050649_

Round 1

Reviewer 1 Report

The authors present a study with a novel approach that may be of wide interest to general audiences. The tables and figures are clear and well presented. However, there are concerns about the clarity of the manuscript particularly in the methods and results sections. There is also missing information which is then later presented as appendices, but not included in the manuscript. 

Comment 1: Please review the manuscript for spelling errors, there are a few still present such as in the first line of the abstract (van instead of can) or in the introduction (line 41 – fosters instead of foster). There are also several grammatical and sentence structure errors.

Comment 2: This is a personal preference, and the authors are free to ignore this suggestion if they wish: Sentences which include e.g. as a transition between ideas are unclear and take more effort to read than if the idea had been laid out as an example to begin with. To clarify, in line 60 the authors state “Cross-cultural differences in narrative comprehension have been found e.g. in character evaluation, plot development and time/space imagination”. This sentence structure makes the idea less clear than simply stating “For example, studies have found that there are cross-cultural differences in narrative comprehension, character evaluation, plot development, and time/space imagination.”

Comment 3: Recent ethical guidelines advise against using the word “subject” to refer to study participants. Please revise.

Comment 4: Were you scanning for 71 minutes continuously? That is an extremely long time to have a participant focus on a task. How are you accounting for potential confounding effects such as fatigue, eye strain, auditory strain from the repetitive noise of the scanning, and muscle strain from laying still for this period of time which may also introduce unwanted motion in your images? How are you managing the fact that some participants asked for breaks and some did not?

Comment 5: In section 2.2 you describe the story as having sections that are associated with either Finnish or Russian cultural backgrounds and stereotypes. How was this assessment made? Who wrote this story? If the authors created this story for the purposes of this study, were there independent raters that categorized certain sections as either relating more to Russian or Finnish culture? Also, which language was the story presented in? You mention in the discussion that all participants had an equal fluency of the Finnish language. How did you measure this?

Comment 6: To what degree do the authors use the IAT in making their claims/interpreting their results? The use of this test in this context is confusing. Reading further into the manuscript, it seems the authors employ an association test, but not the IAT specifically. This is unclear from the methods. Later in the appendix, the IAT itself is described as having been used, but this is not in the methods section of the manuscript. What was the justification for using the IAT considering some of the criticisms of the test?

Comment 7: What time frame was the take-home task performed in. If the authors asked the participants to recall what was on their mind during the brain imaging (and especially considering how long the imaging session was) it is unlikely that participants were able to accurately recall this information. How do the authors account for the potential that participants reported what came to mind at the time of the task, and substituted this for what was on their mind during the imaging?

Comment 8: The methods section is concerningly unclear and is missing critical information. Line 257 is the first time that the contrast mechanism is mentioned. This is after an entire section detailing the fMRI acquisition parameters and preprocessing. The authors employ a InI sequence instead of EPI and justify this in part by stating that this allows for higher spatial resolution, but acquire data with 5mm x 5mm in plane resolution. They also justify this sequence choice in part by stating that this method allows for a higher sampling rate, but it is unclear why this is a primary concern if they are measuring BOLD contrast, where the haemodynamic response is inherently slow to the point where several measurements per second would not provide any advantage.

Comment 9: What was the purpose of the resting state measurement or of repeating the introductory segment of the audiobook?

Comment 10: Section 2.6 is unclear. What are your final measurements and how are they evaluated for statistical significance? You state that you calculate correlations between subjects, but then use t-tests to compare these between groups. Were the between subjects correlations conducted within each group or across all participants as a whole.

Comment 11: Sections 2.7 is also unclear, with similar concerns as section 2.6. It is unclear which participants you are applying these analyses to at which point, and it is also unclear how many tests are caried out, as well as what certain tests measure. For example, the authors state in line 306 that the Wicoxon test was used to evaluate whether the number of associated words differed between the groups, but it is unclear if this means the authors are comparing how many words in total the two groups reported, or how many words were common between participants in the two groups respectively.

Comment 12: More detail is needed in the discussion to justify the claims made. None of the measurements employed in this study can demonstrate a causal relationship of cultural background of participants shaping certain behaviours and brain processes. The authors also do not discuss what advantages their choice of methods had for the presented results.

Comment 13: The manuscript should be expanded with a limitations section.

Comment 14: It is unclear why many of the methods and results are presented after the manuscript as appendices.

Reviewer 2 Report

This article examined whether and how cultural family background may shape the processing of an audiobook in the human brain with fMRI. The results revealed that even relatively subtle between-group differences in cultural background can result in higher similarity within one group compared to the other in how natural speech is interpreted and how it is processed in multiple brain areas. This topic could have the potentials to help overcome potential challenges in mutual understanding between individuals with different cultural backgrounds and social identities in modern multicultural societies.

1) The research should be better motivated. The authors need to provide more explanation why they conducted this study in the introduction part. Are there some theories that can motivate the current research question?

2) Page 2: “Recent methodological advances have made it possible to both record brain activity…”. Here, the authors should introduce the Inter-subject correlation (ISC) analysis specifically. What is ISC and why use it in this study?

3) Page 2: “We hypothesized to find cultural family background specific enhancements of similarity in how different parts of the story are interpreted as well as in similarity of brain hemodynamic activity in brain areas known to be involved in processing of natural speech.” This sentence is so long and difficult to understand. Moreover, the authors should tell why they have such a hypothesis.

4) The fictional audiobook used in this study was custom-written by the author. Is this audiobook related to Finnish and Russian cultures? Did the author evaluate the validation of this stimulus?

Round 2

Reviewer 1 Report

The authors have made extensive changes to the manuscript which, in my opinion, significantly improve the clarity and the legibility of the text. I would like to take the opportunity to commend the authors on the work done to make the manuscript flow more clearly and be significantly more approachable for a reader. The added information has also improved the discussion and interpretation of the results, and while there are some remaining questions the changes to the text lend more credibility to the results and improve confidence in the scientific validity of the study. There are some points left to address that are laid out below, but these are minor adjustments.

In Line 159 of the introduction the disclaimer is confusing. With the way it is worded, it seems as if the authors specifically attempted to find participants who are not representative of people with a Finnish or Russian background. Do you mean that you were simply focused on family background and were not concerned with how representative these participants were compared to native Russians and native Finns?

In the original manuscript, it was unclear whether the authors had conducted the IAT, a free association test, or both, which made the methods and discussion hard to follow. With the added clarification I support the author’s decision to move the IAT to supplementary materials and focus the discussion on the results of the free association.

Although BOLD is a common acronym in fMRI studies, it is still expected to define the acronym for a reader when first used. Please include a definition when introducing the BOLD response analyses in section 2.6.

Lastly, there are still minor spelling errors in certain sentences that should be checked. 

Reviewer 2 Report

I recommend this manuscript for publication.

Author Response

We thank, once again, the Reviewer for the valuable comments we got from the Reviewer at the first revision round. Those comments greatly improved the manuscript.